# Predicting Pharmaceutical Particle Size Distributions Using Kernel Mean Embedding

**DOI:** 10.3390/pharmaceutics12030271

**Published:** 2020-03-16

**Authors:** Daan Van Hauwermeiren, Michiel Stock, Thomas De Beer, Ingmar Nopens

**Affiliations:** 1BIOMATH—Department of data analysis and mathematical modelling, Ghent University, Coupure Links 653, 9000 Gent, Belgium; ingmar.nopens@ugent.be; 2Laboratory of Pharmaceutical Process Analytical Technology—Department of pharmaceutical analysis, Ghent University, Ottergemsesteenweg 460, 9000 Gent, Belgium; Thomas.debeer@ugent.be; 3KERMIT—Department of data analysis and mathematical modelling, Ghent University, Coupure Links 653, 9000 Gent, Belgium; Michiel.stock@ugent.be

**Keywords:** granulation, wet granulation, continuous manufacturing, process modeling, particle size distributions, kernel methods, kernel mean embedding, predictive modeling, data-driven, machine learning

## Abstract

In the pharmaceutical industry, the transition to continuous manufacturing of solid dosage forms is adopted by more and more companies. For these continuous processes, high-quality process models are needed. In pharmaceutical wet granulation, a unit operation in the ConsiGma${}^{\mathrm{TM}}$-25 continuous powder-to-tablet system (GEA Pharma systems, Collette, Wommelgem, Belgium), the product under study presents itself as a collection of particles that differ in shape and size. The measurement of this collection results in a particle size distribution. However, the theoretical basis to describe the physical phenomena leading to changes in this particle size distribution is lacking. It is essential to understand how the particle size distribution changes as a function of the unit operation’s process settings, as it has a profound effect on the behavior of the fluid bed dryer. Therefore, we suggest a data-driven modeling framework that links the machine settings of the wet granulation unit operation and the output distribution of granules. We do this without making any assumptions on the nature of the distributions under study. A simulation of the granule size distribution could act as a soft sensor when in-line measurements are challenging to perform. The method of this work is a two-step procedure: first, the measured distributions are transformed into a high-dimensional feature space, where the relation between the machine settings and the distributions can be learnt. Second, the inverse transformation is performed, allowing an interpretation of the results in the original measurement space. Further, a comparison is made with previous work, which employs a more mechanistic framework for describing the granules. A reliable prediction of the granule size is vital in the assurance of quality in the production line, and is needed in the assessment of upstream (feeding) and downstream (drying, milling, and tableting) issues. Now that a validated data-driven framework for predicting pharmaceutical particle size distributions is available, it can be applied in settings such as model-based experimental design and, due to its fast computation, there is potential in real-time model predictive control.

## 1. Introduction

In pharmaceutical twin-screw wet granulation (TSWG), a dry powder is granulated using a liquid to wet granules. The collection of granules differs in shape and size. Its measurement is called a particle size distribution (PSD). The measurement of this PSD is important to determine the input setting of the TSWG. A fine balance is sought between aggregating the dry powder enough so that its flow properties are improved but not too much so that problems in the next unit operations arise. Too little aggregation results in dry powder that does not have good flow properties and could be blow out to the filters in the fluid bed dryer. Too much aggregation could lead to particles that are too large, needing longer drying times.

For the rapid development of a new formulation on the powder-to-tablet line under study (ConsiGma${}^{\mathrm{TM}}$-25), it is essential to have a predictive model linking the settings of the TSWG to the PSD. One possible approach is to use a population balance model (PBM), where the dynamical changes in particle size are described by making assumptions on how particles can aggregate and break in the TSWG [1].

This work takes a different look at the problem, using a data-driven approach to directly link the TSWG settings and the resulting PSD at the end of the granulator. The benefit this apprroach is that there is no need to make assumptions about the nature of the PSD itself. This is in contrast to the PBM framework, where thorough knowledge of the dynamics of aggregation and breakage of particles is essential. In Section 2, a general overview of the theory used in this work is given before diving into the mathematical descriptions of all the ideas in Section 3. In Section 4, the experimental set up and data collection are described. Section 5 describes the calibration procedure of the data-driven model, for which the results are presented and discussed in Section 6. General conclusions on this work are drawn in Section 7, and Section 8 lists some potential future research topics.

## 2. General Principles

In this section, a high-level overview explaining the general principles of the methodology in this paper is presented to give the reader an overview of the whole approach before diving into the details.

Supervised machine learning models learn a mapping from an arbitrary input to an arbitrary output space. How does one make a predictive model of a distribution of particle size as a function of the machine settings? One possible approach would be to aggregate the information contained in the whole distribution into a mean particle size, or an indication of the size of the largest or smallest particles. Typically, $d_{10}$, $d_{50}$, and $d_{90}$ are used to describe the particle size in experimental papers in this application field, as in the work of Verstraeten et al. [2]. This approach is sensible if the underlying distribution is known. For instance, if the particle distribution is adequately approximated by a Gaussian distribution, then information on the mean particle size and the standard deviation is enough to fully characterize the distribution. However, in this application there is no knowledge of an underlying theoretical distribution to describe particle size. Hence, we need a framework that is agnostic with respect to the potential distribution types and which can link these with the process parameters.

To model distributions, we have to be able to construct manageable numerical representations. For example, we can compute all the moments of the distribution. In the jargon of the machine learning field, this would be called “feature generation” or a “feature map”. Knowing all the moments of the distribution, the distribution itself is completely characterized. It is not convenient to work with an infinite number of moments if we want to link that to process settings.

Fortunately, there is a way to translate a distribution into a point in an implicit feature space in such a way that all information is retained. This procedure is explained in Section 3.2. This translation into a new space does not require the explicit calculation of a large number of features. It expresses all operations in terms of an inner product between pairs of data points in a feature space. These inner products are calculated using a class of functions called kernel functions. This approach of bypassing the generation of a large number of features and only using inner products is the core of a class of algorithms called kernel methods. A brief introduction to learning with kernels is written in Section 3.1. This transformation should be interpreted as a mapping to a mean in that feature space, hence the name of this technique is kernel mean embedding (KME).

Next, this theory is extended to conditional distributions (i.e., a PSD given some TSWG input settings in Section 3.3). If we have a way to deal with conditional distributions, the next logical step is to derive a learning framework—more specifically, structured output prediction. This framework makes it possible to learn the relationship between the mean embedding of a distribution in a feature space and the TSWG settings. To test if this approach can be generalized in the design space, a leave-one-out cross-validation is performed. The details to derive the learning theory are described in Section 3.4.

This work presents a way to manipulate distributions in such a way that all information is maintained and can be put this into a framework where the relation between the process parameters and the distribution can be learnt. The upside is that this framework allows for easy implementation of cross-validation so that the quality of the learnt relation can be assessed. However, the distributions are still expressed in that new feature space, which is not convenient to interpret. Section 3.5 deals with inverse operation from that feature space to a PSD, that is, recovering the function from the embedding.

In summary, the theory makes it possible to learn the (cross-validated) relationship between an input space and distributions and still maintain interpretability by allowing the inverse transformation to a distribution at the end. For a visual cue, this whole procedure is summarized in Figure 1.

## 3. Theoretical Background

In the field of machine learning, the concept of the KME of distributions is a class of non-parametric methods in which a probability distribution is mapped to an element of a recurrent kernel Hilbert space (RKHS). These methods are a generalization of the classical feature mapping in kernel methods, which use individual data points. The learning framework is general in the sense that it can be applied to arbitrary distributions over any space on which a sensible kernel function can be defined. The focus of this work is on representing the distribution as real-valued scalars (i.e., particle sizes). However, various kernels have been proposed for other data types, such as strings, graphs, manifolds, and dynamical systems [3,4].

This part consists of five subsections: first, a general introduction to kernel methods is given. Next, the Hilbert space embedding of marginal and conditional distributions are discussed, which leads to the formulation of a framework for learning on distributional data. Finally, a method to recover distributions from Gaussian RKHS embeddings is discussed. In this work, we only discuss the relevant background information to understand the applications in this work. For an in-depth review of KME, its properties, and its applications, the reader is referred to the review article by Muandet et al. [5]. More details on the backbone of KME: RKHS, learning with kernels, and probability theory can be found in Hofmann et al. [4], Schölkopf et al. [6], Berlinet and Thomas-Agnan [7], respectively.

### 3.1. Learning with Kernels

The classical machine learning algorithms perceptron [8], support vector machine [9], and principal component analysis [10,11] consider the data, $x,x^{\prime }\in X$, with X a non empty set, through their inner product $\langle x,x^{\prime }\rangle$. This inner product can be interpreted as a similarity measure between the elements of X. This class of linear functions may be too restrictive for many applications if more complex relations between input and output data are sought. The core of kernel methods is to replace the inner product $\langle x,x^{\prime }\rangle$ with another (non-linear) similarity measure. As an example, one can explicitly apply a non-linear transformation:(1)$$\begin{array}{cc} \phi :X & \to F\\ x & \mapsto \phi \left(x\right) \end{array}$$
from X to the high-dimensional feature space F and evaluate the inner product in the newly constructed space:(2)$$k\left(x,x^{\prime }\right)={\langle \phi \left(x\right),\phi \left(x^{\prime }\right)\rangle }_{F},$$
where ${\langle \;\cdot \;,\;\cdot \;\rangle }_{F}$ is the inner product of F, ϕ is the feature map, and *k* is the kernel function which defines a non-linear similarity measure between x and $x^{\prime }$. Given a learning algorithm that operates on the data through the inner product $\langle x,x^{\prime }\rangle$, a non-linear extension of the algorithm can be made by substituting $\langle x,x^{\prime }\rangle$ with ${\langle \phi \left(x\right),\phi \left(x^{\prime }\right)\rangle }_{F}$. The principles of the algorithms do not change, only the space in which the algorithms operate. The complexity of the algorithm is controlled by the complexity of the non-linear transformation ϕ. The evaluation of Equation (Equation 2) requires two steps: explicitly constructing the feature maps $\phi \left(x\right)$ and subsequently evaluating the inner product ${\langle \phi \left(x\right),\phi \left(x^{\prime }\right)\rangle }_{F}$. Issues can arise when $\phi \left(x\right)$ defines a transformation to a high-dimensional feature space. However, it is possible to evaluate ${\langle \phi \left(x\right),\phi \left(x^{\prime }\right)\rangle }_{F}$ directly without explicitly constructing the feature maps. This is an essential part of kernel methods, and in the machine learning community this is called “the kernel trick”. A visual aid for this kernel trick is shown in Figure 2.

This kernel trick can only be applied if *k* is positive definite. The positive definite kernel function $k\left(x,x^{\prime }\right)$ is central to the successful application of KME. This kernel function initially arises as a way to perform an inner product $\langle x,x^{\prime }\rangle$ in a high-dimensional feature space H for some data points $x,x^{\prime }\in X$. The collection of all pairwise inner products within the set of data vectors x is called the n×n-Gram or kernel matrix $K_{ij}:=k\left(x_{i},x_{j}\right)$. In general, a symmetric function *k*, is called a positive definite kernel on X if the Gram matrix is positive definite, that is,
(3)$$\sum_{i=1}^{n}\sum_{j=1}^{n}c_{i}c_{j}k\left(x_{i},x_{j}\right)\geq 0,\;\forall x_{i}\in X.$$

Equation (Equation 3) holds for any n∈N, all finite sequences of points $x_{1},\cdots ,x_{n}$ in X and all choices of *n* real-valued coefficients $c_{1},\cdots ,c_{n}\in R$ [12]. The positive definiteness of the kernel guarantees the existence of a dot product space F and a feature map ϕ:X→F such that $k\left(x,x^{\prime }\right)={\langle \phi \left(x\right),\phi \left(x^{\prime }\right)\rangle }_{F}$ [13] without needing to compute ϕ explicitly [6,9,14,15]. Moreover, a positive definite kernel induces a space of functions from X to R called an RKHS H, hence also called a reproducing kernel [13]. An RKHS has two important properties: first, for any x∈X, the function $k\left(x,\;\cdot \;\right):y\mapsto k\left(x,y\right)$ is an element of H. That is, whenever the kernel *k* is used, the feature space F is essentially the RKHS H associated with this kernel and it can be interpreted as a canonical feature map:(4)$$\begin{array}{cc} k:X & \to H\subset R^{X}\\ x & \mapsto k\left(x,\;\cdot \;\right) \end{array}$$
where $R^{X}$ denotes the vector space of functions from X to R. The second property is that an inner product in H satisfies the reproducing property, i.e., for all functions f∈H and x∈X,
(5)$$f\left(x\right)={\langle f,k\left(x,\;\cdot \;\right)\rangle }_{H}.$$

In particular: $k\left(x,x^{\prime }\right)={\langle k\left(x,\;\cdot \;\right),k\left(x^{\prime },\;\cdot \;\right)\rangle }_{H}$. Further details on how $\phi \left(x\right)=k\left(x,\;\cdot \;\right)$ can be derived directly from the kernel *k* can be found in Schölkopf et al. [6]. The kernel used in this work is a Gaussian kernel, which is part of a class of kernels with interesting properties called radial basis functions (RBFs): (6)$$k^{\mathrm{RBF}}\left(x,x^{\prime }\right)=\exp\left(-\frac{{\left\|x-x^{\prime }\right\|}^{2}}{2\sigma^{2}}\right)$$
with σ>0 a bandwidth parameter. For σ→∞, the Gram matrix of this kernel becomes a matrix of ones, for σ→0, it becomes an identity matrix. The former situation implies that all instances are the same, the latter implies that they are all completely unique. The RBF kernel is a stationary kernel: it can be described as a function of the difference of its inputs. The RBF kernel is also called a universal kernel because any smooth function can be represented with a high degree of accuracy, assuming we can find a suitable value of the bandwidth. More details on different classes of kernel functions and their application domains can be found in Genton [16]. For further details on the properties of the RKHS and important theorems such as Mercer’s and Bochner’s theorem, the reader is referred to Muandet et al. [5], Mercer [12], and Bochner [17], respectively.

### 3.2. Hilbert Space Embedding of Marginal Distributions

In KME the concept of a feature map ϕ is extended to the space of probability distributions through the mapping μ which defines the representer in H of any distribution P: (7)$$\begin{array}{cc} \mu :M_{+}^{1}\left(X\right) & \to H\\ P & \mapsto \int_{X}k\left(x,\;\cdot \;\right)dP\left(x\right), \end{array}$$
with $M_{+}^{1}\left(X\right)$ the space of probability measures over a measurable space X [7,18]. The above mapping is the kernel mean embedding that is considered in this work:(8)$$\phi \left(P\right)=\mu_{P}:=E_{X∼P}\left[k\left(X,\;\cdot \;\right)\right]=\int_{X}k\left(x,\;\cdot \;\right)dP\left(x\right).$$

A visual representation of this mean embedding is given in Figure 3. In essence, the distribution P is transformed into an element in the feature space H, which is just an RKHS corresponding to the kernel positive definite kernel *k*. This element (i.e., the mean embedding $\mu_{P}$) is the expected value in that feature space. Since P is a probability density distribution, the expected value can be written as an integral. The derivation, proof, and properties of Equation (Equation 8) can be found in Muandet et al. [5], Berlinet and Thomas-Agnan [7], and Smola et al. [18].

Through Equation (Equation 8), most RKHS methods can therefore be extended to probability measures. When embedding a distribution in another space, it is crucial to understand what information of the distribution is retained by the kernel mean embedding. Consider the class of inhomogeneous polynomial kernels of order p∈N:(9)kpolyx,x′=〈x,x′〉+1p=1+p1〈x,x′〉+p2〈x,x′〉2+⋯+pp〈x,x′〉p.

Polynomial kernels of order *p* allow for learning a *p*-th order polynomial model w.r.t. the features. For our purposes, a polynomial kernel would model the *p* first moments of a distribution when used in KME. For a linear kernel, which is equal to computing the inner product, $\mu_{P}$ equals the first moment of P, whereas the polynomial kernel of order 2 allows the mean map to retain information on both the first and the second moments of P. Generally speaking, the mean map using the inhomogeneous polynomial kernel of order *p* captures information up to the *p*-th moment of P. Other explicit examples for some kernels can be found in Smola et al. [18], Fukumizu et al. [19], Sriperumbudur et al. [20], Gretton et al. [21], Schölkopf et al. [22]. There exists a class of kernel functions known as characteristic kernels for which the kernel mean representation captures all information about the distribution P, with the Gaussian kernel used in this work as an example [23,24]. It follows that the RKHS endowed with the kernel *k* should contain a sufficiently rich class of functions to represent all higher-order moments of P [24]. The map $P\mapsto \mu_{P}$ is injective, implying that ${\left\|\mu_{P}-\mu_{Q}\right\|}_{H}=0$ if and only if P=Q, that is, P and Q are the same distribution. Injectivity of the map $P\mapsto \mu_{P}$ makes the RKHS embedding suitable for regression problems since this map is inherently structurally identifiable (i.e., each element in the feature space corresponds to one unique distribution in the original space). Lastly, it is necessary to point out that, in practice, access to the true distribution P is often lacking, and thereby the mean embedding $\mu_{P}$ cannot be computed. Instead, often only an independent and identically distributed (iid) sample $\left\{x_{1},\ldots ,x_{n}\right\}$ of the distribution is available. The standard estimator ${\hat{\mu }}_{P}$ of the kernel mean $\mu_{P}$ is an empirical average:(10)$${\hat{\mu }}_{P}:=\frac{1}{n}\sum_{i=1}^{n}k\left(x_{i},\;\cdot \;\right),$$
with ${\hat{\mu }}_{P}$ an unbiased estimate of $\mu_{P}$. By the weak law of larger numbers, ${\hat{\mu }}_{P}$ converges to $\mu_{P}$ as n→∞ [25]. In this work, the data should be interpreted as a probability mass distribution associated with the sample X. For example, $\hat{P}:=\frac{1}{n}\sum_{i=1}^{n}\delta_{x_{i}}$, with $\delta_{x}$ the Dirac measure defined for x in X, such that the mean embedding takes the form of a weighted sum of feature vectors:(11)$${\hat{\mu }}_{P}:=\sum_{i=1}^{n}w_{i}k\left(x_{i},\;\cdot \;\right),$$
with $w=\left[w_{i}\right]\in \Delta^{n-1}$, that is, a histogram with weights $w_{i}>0$ and subject to the constraint $\sum_{i}^{n}w_{i}=1$ [26]. A comparison of Equations (Equation 8), (Equation 10), and (Equation 11) is visualized in Figure 4.

In summary, the described framework allows the transformation of marginal distributions into a rich feature space without making assumptions about the underlying distribution (e.g., belonging to a particular class of distributions) by using an extension of previously established kernel methods. By carefully choosing the nature of this transformation, all information on this distribution is retained. Next, it can be proven that this map is identifiable, which makes this representation suitable for regression problems. Finally, compared to density estimation approaches, the kernel mean representation is less prone to the curse of dimensionality [27,28,29].

### 3.3. Hilbert Space Embedding of Conditional Distributions

In the previous subsection, the fundamentals of the mean map for marginal distributions were laid out. In this subsection, the extension of kernel mean embedding to a conditional distribution $P\left(Y|X\right)$ and $P\left(Y|X=x\right)$ for some x∈X is discussed [26,30]. The conditional distribution captures the functional relationship between the two random variables *X* and *Y*. Conditional mean embedding thus extends the capability of kernel mean embedding to model more complex dependence.

Let k:X×X→R and l:Y×Y→R be positive definite kernels for the domains of *X* and *Y*, respectively. The RKHSs associated with these kernels are H and G. The conditional mean embeddings of the conditional distributions $P\left(Y|X\right)$ and $P\left(Y|X=x\right)$ can be written as $U_{Y|X}:H\to G$ and $U_{Y|x}\in G$, such that they satisfy: (12)$$\begin{array}{cc} U_{Y|x} & =E_{Y|x}\left[\varphi \left(Y\right)|X=x\right]=U_{Y|X}k\left(x,\;\cdot \;\right) \end{array}$$(13)$$\begin{array}{cc} E_{Y|x}\left[g\left(Y\right)|X=x\right] & ={\langle g,U_{Y|x}\rangle }_{G},\;\forall g\in G. \end{array}$$

$U_{Y|X}$ is an operator from one RKHS H to the other RKHS G, and $U_{Y|x}$ is an element in G. Equation (Equation 12) states that the conditional mean embedding of the conditional distribution $P\left(Y|X=x\right)$ corresponds to the conditional expectation of the feature map of *Y* given that X=x. The operator $U_{Y|X}$ is the conditioning operation that when applied to $\phi \left(x\right)\in H$ yields the conditional mean embedding $U_{Y|x}$. Equation (13) describes the reproducing property of $U_{Y|x}$, that is, it should be a representer of conditional expectation in G w.r.t. $P\left(Y|X=x\right)$. Using the definition of Song et al. [26,30]: let $C_{XX}:H\to H$ and $C_{XY}:H\to G$ be the covariance operator on *X* and cross-covariance operator from *X* to *Y*, respectively. Then, the conditional mean embedding $U_{Y|X}$ and $U_{Y|x}$ are defined as:(14)$$\begin{array}{cc} U_{Y|X} & :=C_{XY}C_{XX}^{-1} \end{array}$$(15)$$\begin{array}{cc} U_{Y|x} & :=C_{XY}C_{XX}^{-1}k\left(x,\;\cdot \;\right). \end{array}$$

A visual explanation of these concepts can be found in Figure 5.

Further, Fukumizu et al. [23,31] state that if $E_{Y|X}\left[g\left(Y\right)|X=\;\cdot \;\right]\in H$ for any g∈G, then
(16)$$C_{XX}E_{YX}\left[g\left(Y\right)|X=\;\cdot \;\right]=C_{XY}g.$$

For some x∈X, by virtue of the reproducing property, we have that
(17)$$E_{Y|x}\left[g\left(Y\right)|X=x\right]={\langle E_{Y|X}\left[g\left(Y\right)|X\right],k\left(x,\;\cdot \;\right)\rangle }_{H}.$$

Combining Equations (Equation 16) and (Equation 17) and taking the conjugate transpose of $C_{XX}^{-1}C_{XY}$ yields
(18)$$E_{Y|x}\left[g\left(Y\right)|X=x\right]={\langle g,C_{YX}C_{XX}^{-1}k\left(x,\;\cdot \;\right)\rangle }_{G}={\langle g,U_{Y|x}\rangle }_{G}.$$

It is important to note that the operator $C_{YX}C_{XX}^{-1}$ may not exist in the continuous domain because the assumption that $E_{YX}\left[g\left(Y\right)|X=\;\cdot \;\right]\in H,\;\forall g\in G$ may not hold in general [23,30]. To ensure existence, a regularised version of Equation (13) can be used, that is, $C_{YX}{\left(C_{XX}+\lambda I\right)}^{-1}k\left(x,\;\cdot \;\right)$, where λ>0 is a regularization parameter and I is the identity operator in H. Fukumizu et al. [31] showed that under mild conditions, its empirical estimator is a consistent estimator of $E_{Y|X}\left[g\left(Y\right)|X=x\right]$.

In practice, some technical issues arise: since the joint distribution $P\left(X,Y\right)$ is unknown, $C_{XX}$ and $C_{YX}$ cannot be computed directly. The solution is to rely on the iid sample $\left(x_{1},y_{1}\right),\ldots ,\left(x_{n},y_{n}\right)$ from $P\left(X,Y\right)$. Let $\Upsilon :={\left[\phi \left(x_{1}\right),\ldots ,\phi \left(x_{n}\right)\right]}^{T}$ and $\Phi :={\left[\varphi \left(y_{1}\right),\ldots ,\varphi \left(y_{n}\right)\right]}^{T}$ where ϕ:X→H and φ:Y→G are the feature maps associated with the kernels *k* and *l*, respectively. The corresponding Gram matrices are defined as $K=\Upsilon^{T}\Upsilon$ and $L=\Phi^{T}\Phi$. Using the former definitions, the empirical estimator of the conditional mean embedding is given by
(19)$$\begin{array}{cc} {\hat{C}}_{YX}{\left({\hat{C}}_{XX}+\lambda I\right)}^{-1}k\left(x,\;\cdot \;\right) & =\frac{1}{n}\Phi \Upsilon^{T}{\left(\frac{1}{n}\Upsilon \Upsilon^{T}+\lambda I\right)}^{-1}k\left(x,\;\cdot \;\right)\\ \; & =\Phi \Upsilon^{T}{\left(\Upsilon \Upsilon^{T}+n\lambda I\right)}^{-1}k\left(x,\;\cdot \;\right)\\ \; & =\Phi {\left(\Upsilon^{T}\Upsilon +n\lambda I_{n}\right)}^{-1}\Upsilon^{T}k\left(x,\;\cdot \;\right)\\ \; & =\Phi {\left(K+n\lambda I_{n}\right)}^{-1}k_{x}. \end{array}$$

As derived by Song et al. [30], the conditional mean embedding of $\mu_{Y|x}$ can be estimated using
(20)$${\hat{\mu }}_{Y|x}=\Phi {\left(K+n\lambda I_{n}\right)}^{-1}k_{x}.$$

Let ${\hat{\beta }}_{\lambda }:={\left(K+n\lambda I_{n}\right)}^{-1}k_{x}\in R^{n}$, then Equation (Equation 20) can be written as ${\hat{\mu }}_{Y|x}=\Phi {\hat{\beta }}_{\lambda }=\sum_{i=1}^{n}{\left({\hat{\beta }}_{\lambda }\right)}_{i}\varphi \left(y_{i}\right)$. It should be noted that this last equation is in a form similar to Equation (Equation 11). So, in conclusion, *x* determines the weights for the embedding of $P\left(Y|x\right)$.

### 3.4. Learning on Distributional Data

Zhang et al. [32] and Grünewälder et al. [33] observed that the conditional mean embedding has a natural interpretation as a solution to the vector-valued regression problem. Recall that the conditional mean embedding is defined via $E\left[g(Y)|X=x\right]={\langle g,{\hat{\mu }}_{Y|x}\rangle }_{G}$. That is, for every x∈X, ${\hat{\mu }}_{Y|x}$ is a function on Y and thereby defines a mapping from X to G. Furthermore, the empirical estimator in Equation (Equation 20) can be expressed as ${\hat{\mu }}_{Y|x}=\Phi {\left(K+n\lambda I_{n}\right)}^{-1}k_{x}$, which already suggests that the conditional mean embedding is the solution to an underlying regression problem. Given an iid sample $\left(x_{1},z_{1}\right),\ldots ,\left(x_{n},z_{n}\right)\in X\times G$, a vector-valued regression problem can be formulated as:(21)$${\hat{E}}_{\lambda }\left(f\right)=\sum_{i=1}^{n}\left|\right|z_{i}-f\left(x_{i}\right){\left|\right|}_{G}^{2}+{\lambda ||f||}_{H_{\Gamma }}^{2}\;,$$
where G is a Hilbert space, $H_{\Gamma }$ denotes an RKHS of vector-valued functions from X to G and ${\hat{E}}_{\lambda }$ is the error associated with this regression problem [34]. Grünewälder et al. [33] show that ${\hat{\mu }}_{Y|X}$ can be obtained as a minimizer of the optimization of Equation (Equation 21). A natural optimization problem for the conditional mean embedding is to find a function μ:X→G that minimizes an objective. Grünewälder et al. [33] shows that that objective can be bounded from above by a surrogate loss function, which in its empirical counterpart is described as
(22)$${\hat{E}}_{s}\left[\mu \right]=\sum_{i=1}^{n}||l\left(y_{i},\;\cdot \;\right)-\mu \left(x_{i}\right){\left|\right|}_{G}^{2}+{\lambda ||\mu ||}_{H_{\Gamma }}^{2},$$
with an added regularization term to provide a well-posed problem and prevent overfitting. This vector-values regression interpretation of conditional mean embedding has the advantage that a cross-validation procedure for parameter or model selection can be used because the loss function is well-defined. Since the analysis is done under the assumption that G is finite-dimensional, the conditional mean embedding is simply the ridge regression of feature vectors. Given ${\hat{\beta }}_{\lambda }:={\left(K+n\lambda I_{n}\right)}^{-1}k_{x}$, the hat matrix, $H_{\lambda }$, in the ridge regression context is defined as: (23)$$\begin{array}{cc} H_{\lambda }k_{x} & ={\hat{k}}_{x}\\ \; & =\Phi {\hat{\beta }}_{\lambda } \end{array}$$(24)$$\begin{array}{cc} H_{\lambda } & =K{\left(K+\lambda I\right)}^{-1}. \end{array}$$

The estimated conditional embedding using leave-one-out cross-validation (LOOCV) is then defined as [35]:(25)$${\hat{\mu }}_{Y|x}^{\mathrm{LOOCV}}={\left(I-\mathrm{diag}\left(H_{\lambda }\right)\right)}^{-1}\left(H_{\lambda }-\mathrm{diag}\left(H_{\lambda }\right)\right)\Phi ,$$
with $\mathrm{diag}\left(\;\cdot \;\right)$ denoting the diagonal matrix. Note that using Equation (Equation 25), it possible to calculate all LOOCV conditional embeddings at once using matrix multiplications. For the interpretability of the results, the underlying distribution needs to be recovered from the LOOCV conditional mean embedding. This is described in the next paragraph.

### 3.5. Recovering Distributions from RKHS Embeddings

Given a kernel mean embedding $\mu_{P}$, is it possible to recover essential properties of P from $\mu_{P}$? This problem is known in the literature as the distributional pre-image problem [36,37,38]. It is important to note that there is a distinction with the classical pre-image problem, which does not involve probability distributions [5]. In this problem, objects in the input space are sought which correspond with a specific KME in the feature space. In this way, meaningful information of an underlying distribution can be recovered from an estimate of its embedding. Let $P_{\theta }$ be an arbitrary distribution parametrized by θ and let $\mu_{P_{\theta }}$ be its mean embedding in H. $P_{\theta }$ can be found by the following minimization problem
(26)$$\begin{array}{cc} \hat{\theta } & =\underset{\theta \in \Theta }{arg min}{\left\|{\hat{\mu }}_{Y}-\mu_{P_{\theta }}\right\|}_{H}^{2}\\ \; & =\underset{\theta \in \Theta }{arg min}\langle {\hat{\mu }}_{Y},{\hat{\mu }}_{Y}\rangle -2\langle {\hat{\mu }}_{Y},\mu_{P_{\theta }}\rangle +\langle \mu_{P_{\theta }},\mu_{P_{\theta }}\rangle , \end{array}$$
subject to appropriate constraints on the parameter vector θ. Equation (Equation 26) minimizes the maximum mean discrepancy (MMD), which is defined by the idea of representing distances between distributions as distances between mean embeddings of features. Applied to this work, ${\hat{\mu }}_{Y}$ should be interpreted as the estimated conditional embedding using LOOCV, defined by Equation (Equation 25). The term $\langle {\hat{\mu }}_{Y},{\hat{\mu }}_{Y}\rangle$ is only a function of the estimated conditional embedding, thus is constant and is left out of the minimization. Assume that $\mu_{P_{\theta }}=\sum_{i=1}^{n}\alpha_{i}\varphi \left(y_{i}\right)$ for some $\alpha \in \Delta^{n-1}$, or in words: $P_{\theta }$ is a histogram. It follows that $\langle \mu_{P_{\theta }},\mu_{P_{\theta }}\rangle =\alpha^{\prime }\left(L+\lambda I\right)\alpha$ with $L_{ij}=l\left(y_{i},y_{j}\right)$. The addition of a regularizing term λ allows us to cast the optimization as a standard quadratic programming problem. Finally, $\langle {\hat{\mu }}_{Y},\mu_{P_{\theta }}\rangle$ is then equal to the dot product of $P_{\theta }$ and the conditional embedding of Equation (Equation 25). The optimization in Equation (Equation 26) can thus be written as
(27)$$\hat{\alpha }=\underset{\alpha \in \Delta^{n-1}}{arg min}\alpha^{\prime }\left(L+\lambda I\right)\alpha -2\alpha \;\cdot \;{\hat{\mu }}_{Y|x}^{\mathrm{LOOCV}}.$$

Although it is possible to solve Equation (Equation 27) and find a distributional pre-image, it is not clear what kind of information of P this pre-image represents. Kanagawa and Fukumizu [38] considers the recovery of the information of a distribution from an estimate of the kernel mean when the Gaussian RBF kernel on Euclidean space is used. They show that under some situations certain statistics of P can be recovered, namely its moments and measures on intervals, from ${\hat{\mu }}_{P}$, and that the density of P can be estimated from ${\hat{\mu }}_{P}$ without any parametric assumption on P (Kanagawa and Fukumizu [38]; Theorem 2).

## 4. Experimental Set Up and Data Collection

The application field of this work is pharmaceutical manufacturing. More specifically, the data gathered for this study originate from the high-shear TSWG unit operation in the ConsiGma${}^{\mathrm{TM}}$-25 system (GEA Pharma systems, Collette, Wommelgem, Belgium) continuous powder-to-tablet line. A schematic representation of the production line is shown in Figure 6. A more in-depth depiction of the TSWG can be found in Figure 7, displaying the input and output data of the system. The experimental set up of this paper is described in the work of Verstraeten et al. [2]. Here, only a summary with the details relevant to this work is given. For the full details, the reader should refer to the aforementioned paper. The TSWG is comprised of two 25 mm diameter self-wiping, co-rotating screws with a length-to-diameter ratio of 20:1. The preblend and the granulation liquid (demineralized water) are introduced into the system by a gravimetric twin-screw loss-in-weight feeder (KT20, K-Tron Soder, Niederlenz, Switzerland), and two out-of-phase peristaltic pumps located on top of the granulator (Watson Marlow, Cornwall, UK), connected to 1.6
mm nozzles. In this work, the data for the hydrophobic model drug were used. The preblend for this model drug contains 60% (*w/w*) hydrochlorothiazide (UTAG, Almere, The Netherlands), 16% (*w/w*) lactose monohydrate (Lactochem^®^ Regular, DFE Pharma, Goch, Germany), 16% (*w/w*) microcrystalline cellulose (Avicel^®^ PH 101, FMC, Philadelphia, PA, USA), 3% (*w/w*) hydroxypropylcellulose (Klucel^®^ EXF, Ashland, Covington, KY, USA), and 5% (*w/w*) crosscarmellose sodium (Ac-Di-Sol^®^, FMC, Philadelphia, PA, USA). A three-level full-factorial experimental design was used to study the influence of the granulation process parameters screw speed (450, 675, and 900 rpm), material throughput (5, 12.5, and 25 kg/h), and liquid-to-solid ratio (0.3, 0.45, and 0.6). An overview of the process conditions of this experimental design is listed in Table 1. This experiment was performed with a fixed screw configuration: two kneading compartments, each comprised of six kneading elements (length = diameter/6 for each element) with a 60° stagger angle, separated by a conveying element with the same length equal to 1.5 times the diameter. The barrel’s jacket temperature was set at 25 ${}^{\circ }C$. The samples were collected at four locations inside the barrel, however, only the measurements at the end of the granulator are considered in this work. After collection, the samples were oven-dried before the measurement of the PSD and other properties. The size and shape distribution of the collected, oven-dried, granule samples were analyzed using a QICPIC particle size analyzer with WINDOX 5.4.1.0 software (Sympatec, GmbH, Clausthal-Zellerfeld, Germany). The number of bins in the data were taken from previous work on population balance models [1]. They were chosen such that experimental data of the PSD by QICPIC could be loaded swiftly without the need for interpolation. The grid was comprised of a total of 35 bins, logarithmically spaced between 8.46
μm and 6765.36
μm.

## 5. Calibration Procedure

First, the process parameters X are standardized by removing the mean and scaling to unit variance. Next, some hyperparameters need to be defined: the bandwidth parameters σ of the RBF kernels and the regularization parameter λ. For the kernel on the grid of the distributions, σ is chosen via the median heuristic [39]: $\sigma^{2}=\mathrm{median}\left\{∥\log_{10}\left(x_{i}\right)-\log_{10}\left(x_{j}\right){∥}^{2}:i,j=1,\ldots ,n\right\}$. Note that the logarithm of grid values is taken, as the grid spans more than three orders of magnitude. This brings the kernel values more closely together and gives more realistic results. For the kernel on the process parameters, the bandwidth is chosen as $\sigma^{2}=0.1$, which is approximately ${}^{1}\;{/}_{10}$ of the length scale. Finally, for numerical stability reasons (especially for the pre-imaging problem), a bias is added to the diagonal of the Gram matrix of both kernels: 0.05 for *k* and 0.1 for *l*. For a visual cue: the heatmaps of two Gram matrices of kernels are given in Figure 8. The regularization parameter λ is estimated via LOOCV: its value is altered so that the squared error between the mean embedding of the measured distributions and estimated distributions via LOOCV is minimized:(28)$$\lambda =\underset{\lambda }{arg min}{∥\mu_{P(Y|X)}-{\hat{\mu }}_{P(Y|X)}^{\mathrm{LOOCV}}∥}_{G}^{2}.$$

To assess the model quality, three different measures are used. In the following equations, P is the measured distribution, and $\hat{P}$ is the estimated distribution using LOOCV KME. The MMD is calculated as shown before in Equation (Equation 26):(29)$$D_{\mathrm{MMD}}={∥\mu_{P}-\mu_{\hat{P}}∥}_{G}^{2}.$$

The root mean squared error (RMSE), or $L_{2}$-norm, is defined as:(30)$$D_{\mathrm{RMSE}}=∥\alpha -\hat{\alpha }∥,$$
with $\alpha \in \Delta^{n-1}$. Last, the Kullback–Leibler (KL) divergence is calculated as:(31)$$D_{\mathrm{KL}}=\sum_{i=1}^{n}\alpha \log\frac{\alpha_{i}}{\hat{\alpha_{i}}}.$$

## 6. Results and Discussion

In Figure A4, results for four different experiments are visualized: the measured distribution and the predicted distribution using LOOCV. For the two figures on the left, the calibrated distributions using a PBM from Van Hauwermeiren et al. [1] are plotted as well. Calibrated distributions using the PBM are not available for the experiments in the two right figures, so only predicted distributions using KME are plotted there. More figures with results can be found in Appendix A. These figures show that a good prediction can be achieved over a wide range of distribution shapes (monomodal, monomodal with high skewness, and bimodal) without any assumption of the nature of the true underlying distribution. Some peculiarities occur in the predictions: in the first and last bins, the KME model always predicts non-zero values. This might be due to the formulation of the ridge-regression-like problem or incorrect retrieval of the distribution in the pre-imaging problem. A switch to a lasso problem, as described by Grünewälder et al. [33], could potentially alleviate this issue. Alternatively, the entropic regularization in Equation (Equation 27) might not be chosen in an optimal fashion. A closer look into experiments 9, 12, and 14 shows that the KME predicts the location of the peaks and the skewness of the distributions accurately. Experiments 9 and 14 have lower values of MMD, RMSE, and KL compared to the previous work using PBMs. Considering experiment 20, the model correctly ignores the measurement error of slight bimodal distribution. Overall, when ignoring the peculiarities in the first and last bins, the model attains better results than previous work using population balance models (PBMs) [1], while at the same time lowering the computational cost of calibration and validation and the number of required parameters to describe the model. The main difference in the resulting distributions between KME and PBM is the amount of separation between the modes in bimodal distributions. In other words, the calibrated distributions using PBM have more separated peaks, which results in a recurrent underprediction between the modes in the middle of the distribution and overprediction at the modes. In terms of objective function value, the PBM work quantified the distance between simulation and measurement using the root mean squared error. In Van Hauwermeiren et al. [1], the average RMSE value is 0.0803. In this work, the average RMSE value is 0.0876. For the MMD, the average values for the KME are higher than the PBM approach. According to the KL distance, the KME approach gave better results than the PBM work. Note that in the aforementioned work, the published results are calibration, whereas, in this work, one extra level of complexity is added: predictions of unseen data. Thus, the results presented in this work using KME achieve similar goodness-of-fit while trying to solve a more complex problem. An overview of all goodness-of-fit values can be found in Table 2. It should be noted that, to the authors’ best knowledge, this work is the first one to compare PBM with KME.

## 7. Conclusions

The KME of distributions is an interesting data-driven framework to learn relations between a certain input space (in this application, TSWG process settings) and measured distributions. It can be written into a form that allows the description of the problem as a kernel ridge regression problem. Using this framework, kernel mean embeddings of distributions can be predicted given certain inputs. With the kernel pre-image problem, the prediction can be translated from the high-dimensional Hilbert space into its original space. This allows the interpretation and evaluation of the framework in its original space. The benefits of using KME are fast calculation (a couple of seconds for the given problem), analytical short-cuts for LOOCV, high-quality predictions for a wide variety of distribution shapes without making any assumption about those distributions, and a small number of parameters. The model only has five hyperparameters, for which only the regularization parameter λ was estimated.

This work shows an intuitive data-driven approach for which the whole workflow can be written in less than 30 lines of code (see Appendix B). This compactness combined with the fact that only one parameter needs to be estimated makes it an attractive choice for practitioners with limited programming knowledge.

The whole calculation of the LOOCV KME and pre-image problem takes only a couple of seconds. This is in stark contrast with the PBM calculation from previous work [1], where a single calculation of a PBM already takes a couple of seconds. Performing the whole parameter estimation to yield the results that were also shown in the previous section takes orders of magnitude more time than our data-driven approach.

In conclusion, the proposed approach to predict PSDs in TSWG is fast, does not make any assumptions about the shape of the data, and most importantly, yields high-quality cross-validated results.

## 8. Prospects

Further improvements of this work could include an extension to learn relationships from distribution to distribution. In this application field, this could be applied to assessing the effect of a change in pre-blend composition on the resulting PSD at the end of the granulator. Further, instead of working with one unit operation, the whole ConsiGma${}^{\mathrm{TM}}$-25 production line could be studied. The whole transformation of distributions in the unit operations (feeder, blender, granulator, dryer, mill, and tablet press) could be mapped.

The data for this work was gathered using the off-line measurement device QICPIC. It might be interesting to investigate if the model can be trained with a similar predictive power using an in-line particle size measurement device. In that way, the time-consuming preprocessing steps could be bypassed and the results could be gathered more swiftly. If the same experimental design was used, all the data could be gathered in a matter of hours. However, in-line measurement is more prone to noise and has a lower resolution. The effect of this on the model predictions needs to be investigated.

This method is here applied to particle size distributions, but could be extended to other types of distribution-like data. This work focused on a distribution of particle size, but for instance, the moisture distribution in a collection of granules or the hardness of a representative set of final tablets are also possible applications. One other example seems obvious: the prediction of mixtures. In this sense, we could answer questions like “what is the behavior of a mixture of powders starting from the attributes of its components?” This is a hot topic in pharmaceutical manufacturing, as generating an adequate mixture that has the desired properties is mostly done using expert knowledge. The method described in this work could help in creating a model-based design for mixtures. To our best knowledge, we see no hurdles in applying the same methodology to other data-driven problems with distributed data.

## Figures and Tables

**Figure 1 pharmaceutics-12-00271-f001:**
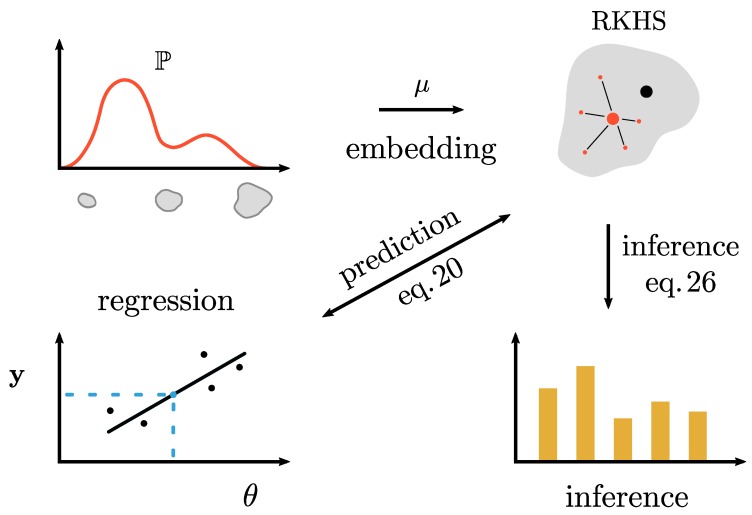
Visual representation of the general modeling principle in this paper. We start at the top left with our data: A measured particle size distribution. This distribution is translated into a new feature space through a kernel function φ. On the top right of the figure, the measured distribution is represented as a mean of features (large red dot), which might be slightly different from the embedding of the true distribution (large black dot). From this embedding we can perform inference or regression with the machine input settings. RKHS: recurrent kernel Hilbert space.

**Figure 2 pharmaceutics-12-00271-f002:**
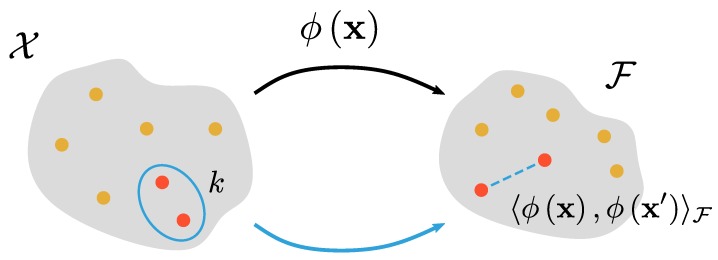
Visual representation of the kernel trick. The value of the kernel function of a pair of objects (denoted in red) in object space X is identical to an inner product of the representations of the objects in the implied Hilbert space F.

**Figure 3 pharmaceutics-12-00271-f003:**
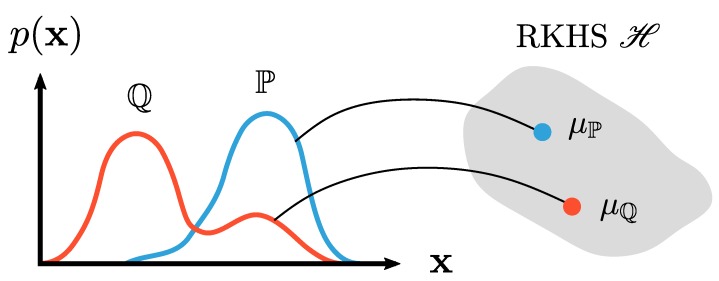
Embedding of marginal distributions P and Q into the RKHS H yielding $\mu_{P}$ and $\mu_{Q}$. Figure based on Muandet et al. [5].

**Figure 4 pharmaceutics-12-00271-f004:**
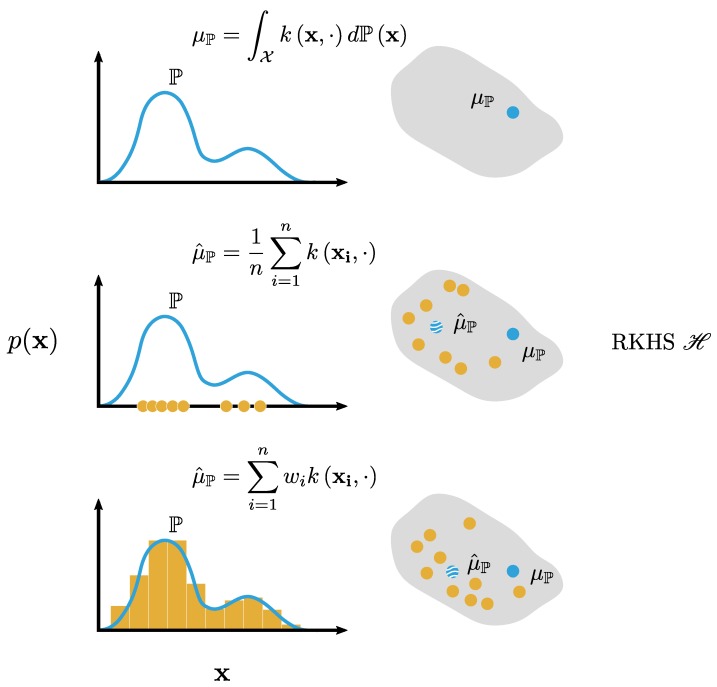
A comparison of the embedding of marginal distributions P into the RKHS H: (**top**) a continuous distribution is an integral over the Hilbert space, (**middle**) a sample distribution is the arithmetic mean over the embeddings of the individual samples, and (**bottom**) a probability mass function is the weighted average over the individual embeddings.

**Figure 5 pharmaceutics-12-00271-f005:**
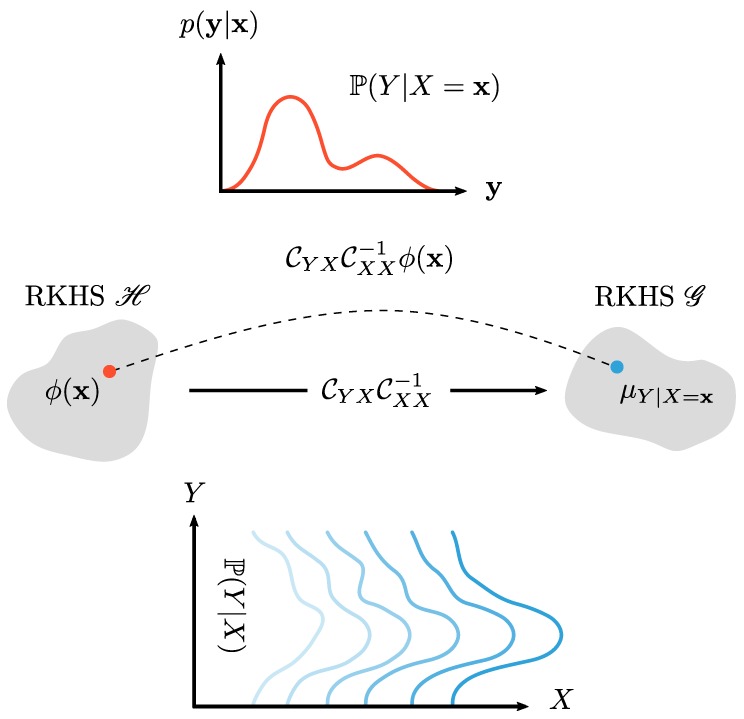
The embedding of conditional distribution P(Y|X) is not a single element in the RKHS. Instead, it may be viewed as a family of Hilbert space embeddings of the conditional distributions P(Y|X=x) indexed by the conditioning variable *X*. In other words, the conditional mean embedding can be viewed as an operator mapping from the RKHS H for features to RKHS G for distributions. Figure based on Muandet et al. [5].

**Figure 6 pharmaceutics-12-00271-f006:**
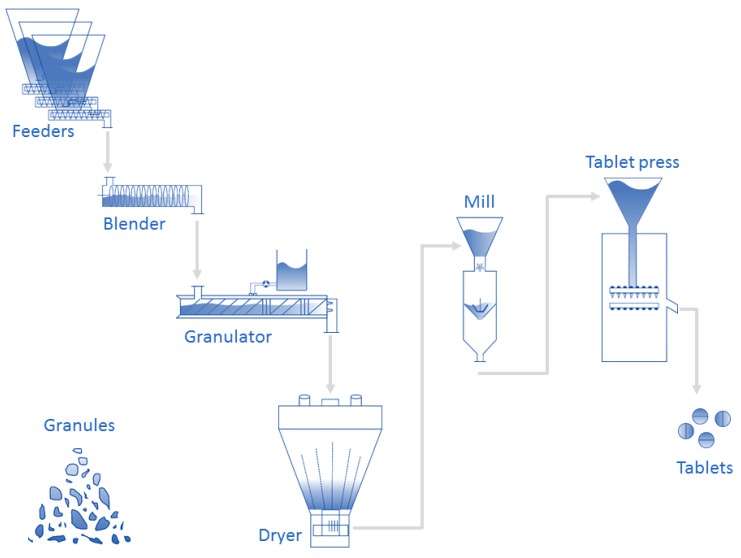
Schematic representation of the ConsiGma${}^{\mathrm{TM}}$-25 system (GEA Pharma systems, Collette, Wommelgem, Belgium) continuous powder-to-tablet line.

**Figure 7 pharmaceutics-12-00271-f007:**
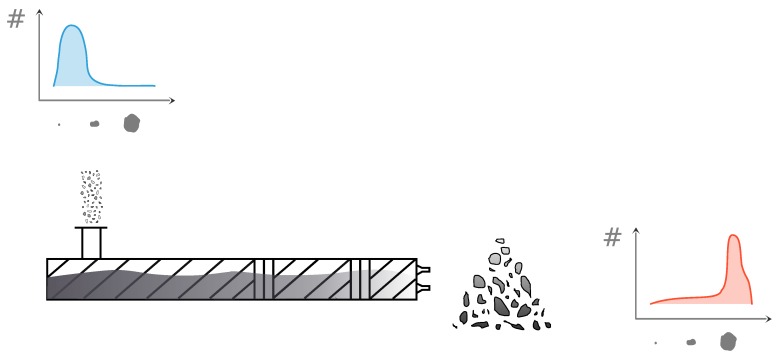
Schematic representation of the input and output data of the twin-screw wet granulation (TSWG), part of the ConsiGma${}^{\mathrm{TM}}$-25 system (GEA Pharma systems, Collette, Wommelgem, Belgium) continuous powder-to-tablet line.

**Figure 8 pharmaceutics-12-00271-f008:**
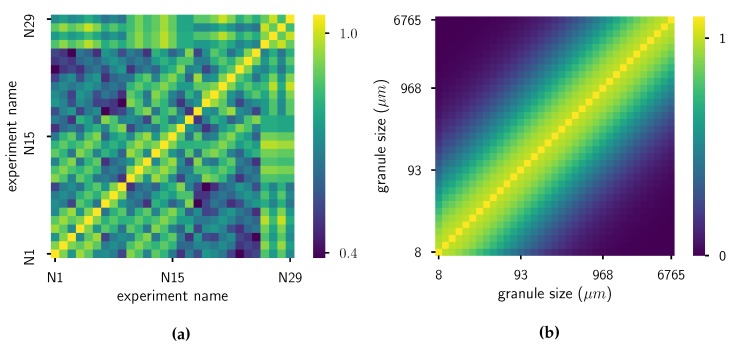
Visualisation of the Gram matrices. (**a**) Gram matrix of kernel *k* on the process settings X. (**b**) Gram matrix of kernel *l* on the grid.

**Figure 9 pharmaceutics-12-00271-f009:**
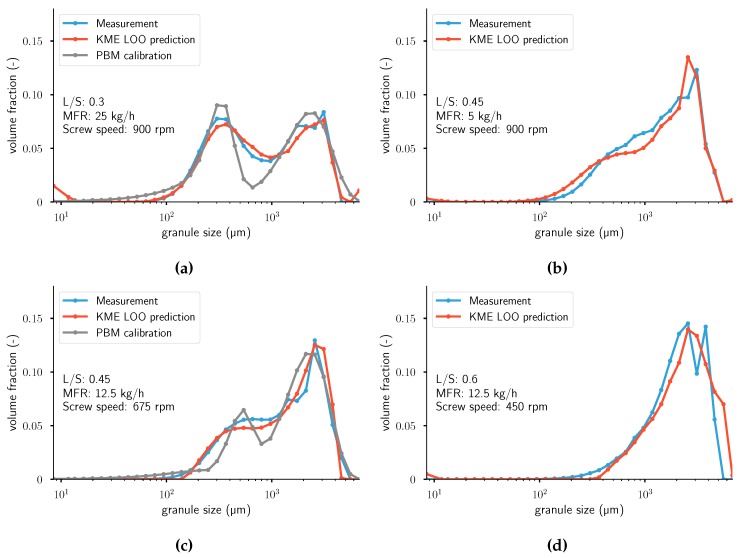
Measurements and leave-one-out cross-validation (LOOCV) predictions of $P_{Y|X}$. For the two
figures on the left, the calibrated distributions using a PBM model from Van Hauwermeiren et al. [1] are plotted as well. From top left to bottom right: (**a**) Experiment 9. (**b**) Experiment 12. (**c**) Experiment 14. (**d**) Experiment 20. Note that the population balance model (PBM) calibration was not performed
for experiments (**b**,**d**). KME: kernel mean embedding; MFR: mass flow rate.

**Table 1 pharmaceutics-12-00271-t001:** Process parameters of the experimental conditions. L/S: liquid-to-solid ratio.

Experiment	Throughput	Screw Speed	L/S Ratio	
	(kg/h)	(rpm)	(—)	
N1	5	450	0	3
N2	5	675	0	3
N3	5	900	0	3
N4	12.5	450	0	3
N5	12.5	675	0	3
N6	12.5	900	0	3
N7	25	450	0	3
N8	25	675	0	3
N9	25	900	0	3
N10	5	450	0	45
N11	5	675	0	45
N12	5	900	0	45
N13	12.5	450	0	45
N14	12.5	675	0	45
N15	12.5	900	0	45
N16	25	900	0	45
N17	25	450	0	45
N18	5	900	0	6
N19	5	450	0	6
N20	12.5	450	0	6
N21	12.5	675	0	6
N22	12.5	900	0	6
N23	25	900	0	6
N24	25	675	0	6
N25	25	450	0	6
N26	12.5	675	0	375
N27	12.5	675	0	525
N28	12.5	675	0	337
N29	12.5	675	0	563

**Table 2 pharmaceutics-12-00271-t002:** Overview of the quality of the LOOCV KME prediction and the calibration of the previous PBM model [1] for each experiment expressed in three distance functions: maximum mean discrepancy (MMD), root mean square error (RMSE), and Kullback–Leibler divergence (KL). At the bottom, the mean of each column is added. The columns of the PBM are missing some values because the calibration could not be performed for those experiments due to missing data in the wetting zone. For more information, see Van Hauwermeiren et al. [1].

Experiment	MMD KME	MMD PBM	RMSE KME	RMSE PBM	KL KME	KL PBM
1	$8.10\times 10^{-4}$	$3.28\times 10^{-3}$	$5.05\times 10^{-2}$	$5.92\times 10^{-2}$	$3.82\times 10^{-1}$	$3.02\times 10^{-1}$
2	$4.73\times 10^{-3}$		$6.22\times 10^{-2}$		$5.44\times 10^{-1}$		
3	$9.59\times 10^{-3}$	$1.57\times 10^{-3}$	$1.00\times 10^{-1}$	$7.49\times 10^{-2}$	$1.84\times 10^{0}$	$7.61\times 10^{-1}$
4	$5.22\times 10^{-3}$		$5.06\times 10^{-2}$		$5.99\times 10^{-1}$		
5	$2.28\times 10^{-3}$		$6.10\times 10^{-2}$		$3.10\times 10^{-2}$		
6	$1.94\times 10^{-4}$		$2.48\times 10^{-2}$		$1.36\times 10^{-1}$		
7	$7.04\times 10^{-3}$	$8.05\times 10^{-4}$	$9.56\times 10^{-2}$	$7.41\times 10^{-2}$	$1.93\times 10^{0}$	$1.80\times 10^{0}$
8	$3.10\times 10^{-3}$		$4.42\times 10^{-2}$		$3.30\times 10^{-2}$		
9	$7.58\times 10^{-4}$	$4.26\times 10^{-3}$	$3.16\times 10^{-2}$	$6.49\times 10^{-2}$	$5.87\times 10^{-1}$	$1.04\times 10^{0}$
10	$2.10\times 10^{-3}$		$7.78\times 10^{-2}$		$2.30\times 10^{0}$		
11	$4.39\times 10^{-3}$		$5.78\times 10^{-2}$		$5.15\times 10^{-1}$		
12	$1.16\times 10^{-3}$		$5.04\times 10^{-2}$		$1.06\times 10^{-1}$		
13	$4.02\times 10^{-2}$		$1.16\times 10^{-1}$		$4.72\times 10^{-1}$		
14	$8.60\times 10^{-4}$	$2.71\times 10^{-3}$	$4.62\times 10^{-2}$	$6.50\times 10^{-2}$	$3.60\times 10^{-2}$	$2.28\times 10^{-1}$
15	$5.46\times 10^{-4}$		$6.22\times 10^{-2}$		$4.22\times 10^{-1}$		
16	$1.15\times 10^{-2}$		$7.74\times 10^{-2}$		$4.76\times 10^{-1}$		
17	$5.27\times 10^{-2}$		$1.55\times 10^{-1}$		$3.53\times 10^{0}$		
18	$3.73\times 10^{-2}$	$7.26\times 10^{-4}$	$1.63\times 10^{-1}$	$1.10\times 10^{-1}$	$9.09\times 10^{-1}$	$1.92\times 10^{0}$
19	$1.39\times 10^{-3}$	$1.07\times 10^{-3}$	$6.00\times 10^{-2}$	$5.76\times 10^{-2}$	$9.35\times 10^{-1}$	$7.98\times 10^{-1}$
20	$4.90\times 10^{-3}$		$9.76\times 10^{-2}$		$2.34\times 10^{0}$		
21	$6.51\times 10^{-3}$		$8.01\times 10^{-2}$		$5.39\times 10^{-1}$		
22	$4.99\times 10^{-2}$		$1.55\times 10^{-1}$		$4.45\times 10^{-1}$		
23	$4.58\times 10^{-2}$	$1.52\times 10^{-3}$	$1.38\times 10^{-1}$	$6.83\times 10^{-2}$	$9.43\times 10^{-1}$	$7.34\times 10^{-1}$
24	$6.31\times 10^{-3}$		$9.74\times 10^{-2}$		$9.75\times 10^{-2}$		
25	$5.53\times 10^{-2}$	$3.52\times 10^{-3}$	$3.01\times 10^{-1}$	$1.48\times 10^{-1}$	$9.82\times 10^{-1}$	$3.21\times 10^{0}$
26	$2.61\times 10^{-3}$		$3.70\times 10^{-2}$		$3.12\times 10^{-2}$		
27	$8.45\times 10^{-3}$		$1.16\times 10^{-1}$		$7.64\times 10^{-2}$		
28	$8.80\times 10^{-3}$		$6.52\times 10^{-2}$		$5.45\times 10^{-2}$		
29	$4.31\times 10^{-3}$		$6.64\times 10^{-2}$		$9.46\times 10^{-2}$		
mean	$1.31\times 10^{-2}$	$2.16\times 10^{-3}$	$8.76\times 10^{-2}$	$8.03\times 10^{-2}$	$7.38\times 10^{-1}$	$1.20\times 10^{0}$

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
