# Peer review of "Predicting Pharmaceutical Particle Size Distributions Using Kernel Mean Embedding"

_pharmaceutics, 2020, doi:10.3390/pharmaceutics12030271_

Round 1

Reviewer 1 Report

The manuscript entitled "Predicting Pharmaceutical Particle Size Distributions using Kernel Mean Embedding" provides a data-driven mathematical model that links the wet granulation settings with the output distribution of granules. This work is high relevant as it can be easily translated to other distribution-like processes in the pharmaceutical industry for implementation of a rational quality by design approach.

The manuscript magisterially provides wide theoretical background on the concept of KME of distributions. The added value of the utility of the mathematical model to predict distribution shapes without any assumption on the nature of the true underlying distribution could be greater highlighted. The results and discussion section could be expanded with discussion of particular values shown in Table 2, as it would enlighten its content for journal readership. Moreover, some abbreviations should be defined at first appeareance (i.e wrt, but some more may need a quick revision).

Author Response

The results and discussion section are expanded with some examples. The paper is proofread again: some typos and unclear abbreviations are corrected. The discussion of some particular values of the objective function is added for the discussed experiments (line 361-362) as well as for the mean of the distance function values (line 373 - 374)

Reviewer 2 Report

The authors have proposed a machine learning methodology to treat the particle size data and wet granulation process into a Kernel Hilbert Space, where the relationships between process parameters and particle size of the granules may be explored. This method of transforming the data into high-dimension are highly relevant to the current data engineering/analysis in computer science and it is extremely encouraging to see its application for the pharmaceutical process understanding. The development of advanced simulation tools such like this should be highly commendable for the future of pharmaceutical manufacturing.

Just few comments/questions:

Parsum inline granular particle size measurement has been widely used for fluid bed drying, can this be used as direct particle size measurement in a TSG setting? Will the particle size of granules (TSG) be significant on the final quality of the tableting process, since the granules will be then dried and milled for tableting? Since this methodology can be applied for any distribution-like data, how about the quality attributes of the final tablets? Author explained that this method can be used to establish a model based prediction for the design of the formulation, what would be the difference/advantages between this and PLS modelling that is commonly used in a wide range of DoE software?

Author Response

Parsum inline granular particle size measurement has been widely used for fluid bed drying, can this be used as direct particle size measurement in a TSG setting?

The use of Parsum in the TSWG has been investigated by a couple of colleagues in the application of online measurements and control. For a full description, please consult chapter 9 in the thesis of Niels Nicolai. The downside is that it is not trivial to couple Parsum with the TSWG. An interfacing device is needed to not over- or underload the device. Over- or underloading will result in low-quality measurements. Further, Parsum has a lower resolution, i.e. the number of bins, than the tabletop QICPIC device which was used for the measurements in this work. It is not mentioned in the work of Nicolai, but based on expert knowledge on the measurement device we have, the distribution could be only measured for a maximum of 11 bins. This lower resolution would seriously hamper any modelling effort. Therefore it was chosen to perform the modelling work with offline data. There is work ongoing on inline measurement and control using a Parsum device, and we hope to couple the findings of this work with the Parsum data in the future. I have added a small remark in the prospects regarding the use of an inline measurement device (lines 410-418).

reference: Nicolaï, N. (2019). Supervisory process monitoring, identification and control for continuous pharmaceutical wet granulation. Universiteit Gent.

Will the particle size of granules (TSG) be significant on the final quality of the tableting process, since the granules will be then dried and milled for tableting?

The particle size will certainly change in the next unit operations: breakage in the fluid bed dryer and the mill. The particle size is important for the dryer: too large particles will result in longer drying times and too small particles will be blown out into the filter. The main reason for this modelling effort should be seen as this is a first step for a generic data-driven model of the particle size in TSG. Using this model, we can simulate the particle size for new process settings, and in the future simulate the effect of blend properties on the particle size. In this way, we can determine the optimal granulation setting to attain a certain particle size distribution. This would give a huge advantage if a new formulation is tested for production: less experimental runs and being able to find the optimal experimental settings faster will result in a faster time-to-market. The effect of the particle size distribution on the final quality of tablets is not straightforward to determine as the formulation properties play a huge role. In previous work on a flowsheet model of the whole powder-to-tablet line, the quality of the resulting tablets was simulated using the bulk and tapped density. These densities are affected by the particle size distribution.

reference: Metta, N., Ghijs, M., Schäfer, E., Kumar, A., Cappuyns, P., Van Assche, I., Singh, R., et al. (2019). Dynamic flowsheet model development and sensitivity analysis of a continuous pharmaceutical tablet manufacturing process using the wet granulation route. PROCESSES, 7(4).

Since this methodology can be applied for any distribution-like data, how about the quality attributes of the final tablets?

If the aim is to predict the distribution of e.g. hardness in final tablets than this framework could be applied as well. Some remarks on this are included in the prospects (line 417-419 and line 423-425)

Author explained that this method can be used to establish a model based prediction for the design of the formulation, what would be the difference/advantages between this and PLS modelling that is commonly used in a wide range of DoE software?

PLS modelling of TSG quality attributes related to the particle size is based on d10, d50, d90 and the span (ADD SOME REFS). If you are 100% certain about the shape of your resulting distribution, e.g. imagine a particle size distribution which approximates a normal or lognormal distribution, then this approach is justified. However, if you are dealing with distributions in a variety of different shapes for which you do not know the underlying theoretical distribution, it is essential that you use a modelling framework that does not make any assumptions on the data. Therefore, we think that this approach using kernel mean embedding is both justified and widely applicable for particle size distributions. These remarks are added to the general principles section (lines 51-63).
If the aim is to simulate for instance friability, bulk or tapped density, which are properties of the mixture as a whole, than KME is not the correct framework to be used. A PLS model could then for instance be used to link formulation properties and friability.

Reviewer 3 Report

The submission entitled "Predicting Pharmaceutical Particle Size Distributions using Kernel Mean Embedding" is a technically sound, mathematically rigorous and industrially relevant downstream processing case study, characterized by elaborate methodological descriptions and a clear (perhaps very narrow, but well justified) focus on a particular unit operation and equipment type/configuration, which is evaluated with great attention to detail. 

The manuscript is of high quality, but there are some clear shortcomings which must be urgently addressed before it is made publishable:

  1. The Introduction seems to have nothing to do with the focus of the paper, as it starts from the deep end of machine learning, providing certainly rigorous mathematical/statistical/methodological details, but without any motivation. It is a lot more important to gently and convincingly introduce the reader to the current (industrial!) problem that is to be served, to review current progress (and alternative/previous/less promising methods!) of addressing PSD prediction in pharma, to pinpoint the shortcomings, and THEN to introduce the new methodology and cover its algorithmic specifics. A lot more (esp. in terms of Lit. Refs.) should be offered to the reader for reference/comparison.
  2. The KME method has been analyzed diligently and implemented convincingly. Nevertheless, the comparisons vs. PBM approaches are lacking. It is of importance to also articulate this alternative and widespread (PBM) method, and describe its specific implementation better (how was it done here?). The discrepancies in terms of additional peaks/mismatch are clear and well discussed, but is this inevitable, or has it resulted from code input/parameter choices/arbitrary decisions of the authors during PBM implementation? Why is it that PBM calibration hasn't worked? Was it properly attempted? Is there evidence/published precedents of method comparisons (KME vs PBM)?
  3. The Conclusions section appears disproportionately short, and does not convey adequate messages to the practitioner. How can this method be applied, under what constraints and which (data/sensor) cost? Are there limits of applicability/fidelity? What would you recommend for its faster/easier/wider adoption? Please share your thoughts in detail (even better if discussed vs. recent Lit. Ref. precedents).
  4. The standard of english language (grammar, syntax, vocabulary) used is adequately high, but there are several typos scattered throughout the manuscript (e.g. Abstract/manufactering), so a thorough and repeated proofreading is strongly recommended to ensure language clarity.

 Overall, this is a technically strong contribution and a great effort; kindly please improve it to ensure wider impact...

Author Response

The introduction seems to have nothing to do with the focus of the paper, as it starts from the deep end of machine learning, providing certainly rigorous mathematical/statistical/methodological details, but without any motivation. It is a lot more important to gently and convincingly introduce the reader to the current (industrial!) problem that is to be served, to review current progress (and alternative/previous/less promising methods!) of addressing PSD prediction in pharma, to pinpoint the shortcomings, and THEN to introduce the new methodology and cover its algorithmic specifics. A lot more (esp. in terms of Lit. Refs.) should be offered to the reader for reference/comparison.

The introduction has been updated to start from the need of modelling particle size distributions. A reference to alternative approaches such as PBM is included.
A new section is added called "General principles". In that section, a high-level overview of the theory is given so that the reader can first look at a global overview before diving into the details of the theory. To make it extra clear for non-modelling experts, a general overview figure is added as well (figure 1)

The KME method has been analyzed diligently and implemented convincingly. Nevertheless, the comparisons vs. PBM approaches are lacking. It is of importance to also articulate this alternative and widespread (PBM) method, and describe its specific implementation better (how was it done here?). The discrepancies in terms of additional peaks/mismatch are clear and well discussed, but is this inevitable, or has it resulted from code input/parameter choices/arbitrary decisions of the authors during PBM implementation? Why is it that PBM calibration hasn't worked? Was it properly attempted? Is there evidence/published precedents of method comparisons (KME vs PBM)?

It was chosen not to describe the PBM method in detail, as the first author has published the PBM work which is used to compare it to the data-driven method. The same data is used, so it was deemed an unnecessary repetition of already available information.
The PBM modelling effort in previous work was performed using a good modelling practice. There is no reason to assume that the calibration efforts were not properly attempted. It should be noted, that the governing equations for aggregation and breakage are of course an assumption of what could take place. A mismatch could be present, however, this is deemed unlikely. Last, it should be noted that a follow up of the previous PBM paper is underway where issues such as calculating distances between two PBM simulations, identifiability of the aggregation and breakage models, and the inclusion of measurement error are discussed.
There are no precedents of method comparisons (PBM vs KME). This point is added to the discussion section (line 378-379).

The Conclusions section appears disproportionately short, and does not convey adequate messages to the practitioner. How can this method be applied, under what constraints and which (data/sensor) cost? Are there limits of applicability/fidelity? What would you recommend for its faster/easier/wider adoption? Please share your thoughts in detail (even better if discussed vs. recent Lit. Ref. precedents).

I have added sections in the conclusions and the prospects to answer these questions. I have added suggestions on how this work could be extended to online data (lines 410-415). To our best knowledge, we see no limits in the applicability to any type of distributed data. In our case, we have used this for particle size, but other applications could be the distribution of moisture content in the granules or the distributions of hardness in the final tablets (lines 417-423). In the submitted work, I have already added all the code needed to calculate the whole model described in this work (lines 448 - 503). In that way, by giving the reader all the necessary tools, we hope to see this approach rapidly applied to other formulations/applications.

The standard of english language (grammar, syntax, vocabulary) used is adequately high, but there are several typos scattered throughout the manuscript (e.g. Abstract/manufactering), so a thorough and repeated proofreading is strongly recommended to ensure language clarity.

The work was proofread and several typos were updated.